# DuMA: Dual Matching Aggregation for Image-to-Point Cloud Registration

## Abstract

Aligning 2D images with 3D point clouds remains a challenging problem due to intrinsic modality differences. In this paper, we introduce Dual-view Matching Aggregation (DuMA), a novel image-to-point cloud registration framework designed to address this challenge. Our approach incorporates a dual-view matching strategy that harmonizes 2D-3D and 3D-3D correspondences, leveraging complementary insights from both modalities. We design a score aggregation module that fuses dual correspondence scores through a detailed analysis of neighborhood relationships, thereby inducing a robust geometric verification effect and enforcing spatial consistency. To reduce the burden associated with high-dimensional score aggregation, we additionally propose an innovative Anchor-Pivot 5D encoder that decomposes and processes multi-modality scores. Extensive experiments on challenging indoor and outdoor datasets demonstrate that our method significantly mitigates ambiguity while delivering robustness and effectiveness in complex scenes. Code and models will be made available: TBD.

## 1 Introduction

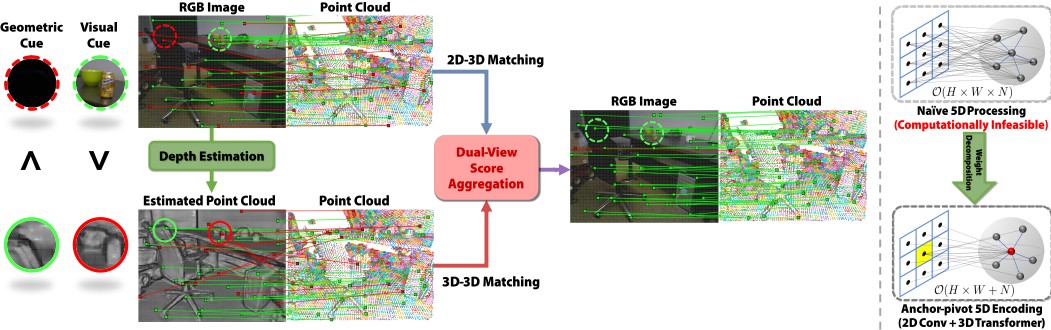

Figure 1: (a) Our proposed DuMA framework integrates both cross-modal (2D–3D) and intra-modal (3D–3D) matching through a dual-view score aggregation process. By capturing complementary cues from image and point cloud data, DuMA enhances alignment accuracy and robustness in challenging scenes. (b) To aggregate high-dimensional multi-modality matching scores, we introduce an Anchor-Pivot 5D encoder that employs a decomposition technique to significantly reduce the computational overhead associated with high-dimensional operations.

Image-to-Point Cloud (I2P) registration is crucial in many computer vision applications that require precise pixel-to-point correspondences, such as Simultaneous Localization and Mapping (SLAM), Augmented Reality (AR), 3D reconstruction, and visual localization.

Achieving accurate registration between 2D images and 3D point clouds is inherently challenging due to the distinct nature of these modalities. Traditional 2D–3D matching approaches Wang et al. (2021); Li et al. (2023); Feng et al. (2019); Pham et al. (2020); Wu et al. (2024) face fundamental difficulties: while 2D images provide rich visual cues, such as color and texture, 3D point clouds primarily encode spatial geometry, making direct correspondence non-trivial. This disparity between

visual and spatial information can lead to ambiguities and inaccuracies, particularly in complex or cluttered scenes, which ultimately affects registration reliability.

Recent efforts have aimed to bridge the gap between image and point cloud modalities by creating unified representations for robust correspondence estimation. For example, FreeReg Wang et al. (2024) fuses RGB and depth features into a shared modality to facilitate correspondence estimation. However, its fully non-trainable design, coupled with the lack of explicit 2D-3D feature interactions and joint optimization, limits its adaptability in complex or ambiguous scenes.

In this paper, we introduce DuMA, a novel dual-view matching aggregation registration framework for image-to-point cloud alignment. Aligning 2D images with 3D point clouds is challenging due to inherent modality differences, and our approach is designed to address this issue by harmonizing both 2D–3D and 3D–3D correspondences. Figure 1(a) illustrates why harmonizing the two correspondence types is essential. The 2D–3D matches rely on visual cues, so they excel in texture-rich regions but generate false matches where colors are similar (e.g., the top of the chair). In contrast, 3D–3D matches depend solely on geometry and therefore capture shape-distinct areas accurately, yet struggle on repetitive structures lacking distinctive visual information. Thus, DuMA extracts complementary cues by matching features across both views, thereby enhancing cross-modality alignment.

To further boost matching reliability, we design a score aggregation module that fuses dual correspondence scores through a detailed analysis of neighborhood relationships, inducing a robust geometric verification effect and enforcing spatial consistency. Unlike traditional methods that rely solely on feature similarity, our module leverages spatial relationships and geometric constraints to filter out ambiguous or incorrect matches. By aligning feature representations with their underlying geometric properties, this approach significantly reduces false correspondences and improves registration robustness, especially in complex or cluttered environments.

A major challenge in multi-modal registration is the computational burden associated with high-dimensional score aggregation. As shown in Figure 1(b), considering both 2D and 3D spatial dimensions simultaneously can lead to prohibitive complexity, making a naive 5D convolution virtually impossible in practice. To overcome this, we propose an innovative Anchor-Pivot 5D encoder that decomposes high-dimensional matching scores into separate 2D and 3D components. This decomposition not only reduces computational overhead but also preserves robust alignment.

Extensive experiments on indoor and outdoor datasets demonstrate that DuMA significantly mitigates ambiguity while achieving state-of-the-art performance in terms of inlier ratio, feature matching recall, and registration recall.

Our key contributions can be summarized as follows:

- We present DuMA, a novel image-to-point cloud registration framework that harmonizes 2D–3D and 3D–3D correspondences for robust multi-modal alignment.
- To enhance matching reliability, we design a score aggregation module that fuses dual correspondence scores through detailed neighborhood analysis and geometric verification.
- We develop an innovative Anchor-Pivot 5D encoder that decomposes high-dimensional matching scores into separate 2D and 3D components, reducing computational overhead.
- With the aforementioned contributions, DuMA achieves state-of-the-art performance on several image-to-point cloud registration benchmarks on both indoor and outdoor datasets.

## 2 RELATED WORK

### 2.1 CORRESPONDENCE-BASED REGISTRATION.

Correspondence-based methods estimate feature correspondences and recover the relative transformation using robust pose estimators. Classical approaches relied on handcrafted features Dalal & Triggs (2005); Lowe (2004); Bay (2006), while recent works leverage deep learning for improved matching in both 2D Lee et al. (2021); Cho et al. (2021); Kim et al. (2022); Huang et al. (2022); Tang et al. (2023); Li et al. (2024) and 3D Yu et al. (2021); Choy et al. (2019); Qin et al. (2023); Huang et al. (2021); Yu et al. (2023a;b); Chen et al. (2023) registration. However, adapting these single-modality techniques to image-to-point registration requires modality conversion, which leads to information loss and degraded performance.

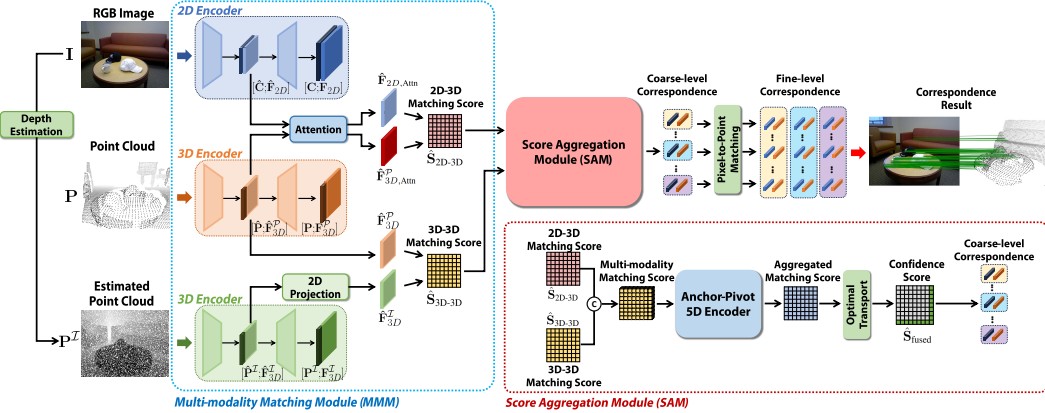

Figure 2: Overall Architecture of the proposed DuMA. DuMA consists of three main parts: Multi-Modality Matching Module, Score Aggregation Module, and fine correspondence matching.

## 2.2 IMAGE-TO-POINT CLOUD REGISTRATION.

Image-to-point cloud registration aims to bridge the modality gap and establish reliable correspondences between images and 3D point clouds. Previous works have addressed this by generating points from images to enable intra-modality comparisons Shotton et al. (2013); Brachmann & Rother (2019); Li et al. (2020), or by extracting and matching keypoints across modalities Feng et al. (2019); Pham et al. (2020); Wang et al. (2021). Recently, methods such as coarse-to-fine matching with multi-scale patches Li et al. (2023), diffusion model-based progressive refinement Wu et al. (2024); Mu et al. (2025), and channel-adaptive feature enhancement Cheng et al. (2025) have significantly improved registration performance. FreeReg Wang et al. (2024) unifies RGB and depth features to close the modality gap, but lacks explicit modeling of inter-modality feature correlations and geometric consistency verification, resulting in ambiguities in challenging scenes. To address these limitations, we propose a novel 5D anchor-pivot encoder that explicitly integrates 2D-3D feature interactions with joint optimization, thereby enhancing geometric consistency and matching robustness.

## 3 METHOD

### 3.1 OVERVIEW

Our proposed method first establishes correspondences at the 2D patch and 3D cluster, then determines pixel-to-point correspondences within each matched 2D patch–3D cluster pair. To this end, we propose two modules: the Multi-modality Matching Module (MMM) and the Score Aggregation Module (SAM). Our Multi-modality Matching Module (MMM) extracts 2D and 3D features from images and 3D features from point clouds. Subsequently, Our Score Aggregation Module (SAM) takes matching scores from 2D-3D and 3D-3D matching as input, and aggregates these scores into a single matching score, while considering neighboring regions' scores to enhance alignment accuracy. The overall architecture is depicted in Figure 2.

### 3.2 PROBLEM STATEMENT

Given a 2D image $\mathbf{I} \in \mathbb{R}^{H \times W \times 3}$ and a 3D point cloud $\mathbf{P} \in \mathbb{R}^{N \times 3}$, the task of 2D-3D registration is to determine the transformation $\mathbf{T}$, defined by a rotation $\mathbf{R} \in \mathrm{SO}(3)$ and a translation $t \in \mathbb{R}^3$. By establishing correspondences $\mathcal{C} = \{(x_i, y_i) \mid x_i \in \mathbb{R}^3, y_i \in \mathbb{R}^2\}$ between 3D points and 2D pixels, the transform can be solved by:

$$\min_{R,t} \sum_{(x_i, y_i) \in C} \|Proj(Rx_i + t, K) - y_i\|^2, \tag{1}$$

where $K$ denotes the intrinsic parameters of the camera, and $Proj(\cdot, \cdot)$ is the function projecting 3D points onto the 2D image plane. Our focus is on refining the correspondence estimation process,

as the precision of these correspondences plays a pivotal role in the accuracy and stability of the resulting alignment transformation. The predicted correspondences can be leveraged to estimate the transformation matrix using PNP-RANSAC Lepetit et al. (2009).

### 3.3 Multi-modality Matching Module (MMM)

The MMM module extracts features from an image $\mathbf{I}$ and a point cloud $\mathbf{P}$ at both coarse and fine levels. Specifically, from an image $\mathbf{I}$, MMM extracts both 2D and 3D features at each level, resulting in four distinct features. From a point cloud $\mathbf{P}$, it extracts only 3D features at both levels, resulting in two distinct features. In total, MMM outputs six unique features (four for the image and two for the point cloud). The six features are (1) $\hat{\mathbf{F}}_{2D} \in \mathbb{R}^{\hat{H} \times \hat{W} \times \hat{d}}$, (2) $\mathbf{F}_{2D} \in \mathbb{R}^{H \times W \times d}$, (3) $\hat{\mathbf{F}}_{3D}^{\mathcal{P}} \in \mathbb{R}^{\hat{N} \times \hat{d}}$, (4) $\mathbf{F}_{3D}^{\mathcal{P}} \in \mathbb{R}^{N \times d}$, (5) $\hat{\mathbf{F}}_{3D}^{\mathcal{I}} \in \mathbb{R}^{\hat{H} \times \hat{W} \times \hat{d}}$, (6) $\mathbf{F}_{3D}^{\mathcal{I}} \in \mathbb{R}^{H \times W \times d}$.

In the above notation, the hatted character $(\hat{\cdot})$ represents the features at the coarse level, while the vanilla character denotes the features at the fine level. This module is designed to not only perform 2D-3D and 3D-3D matching but also to jointly learn and integrate their complementary geometric information.

**2D Backbone.** Following Wu et al. (2024), let $\hat{\mathbf{F}}_{2D} \in \mathbb{R}^{\hat{H} \times \hat{W} \times \hat{d}}$ and $\mathbf{F}_{2D} \in \mathbb{R}^{H \times W \times d}$ represent the 2D features extracted from the image using 2D backbones such as ResNet He et al. (2016) and FPN Lin et al. (2017). $\hat{\mathbf{F}}_{2D}$ is the feature down-sampled at the patch level (coarse level), whereas $\mathbf{F}_{2D}$ is the feature obtained at the pixel level (fine level). We denote the corresponding coordinate matrices of $\hat{\mathbf{F}}_{2D}$ and $\mathbf{F}_{2D}$ as $\hat{\mathbf{C}} \in \mathbb{R}^{\hat{H} \times \hat{W} \times 2}$ and $\mathbf{C} \in \mathbb{R}^{H \times W \times 2}$, respectively. In addition, we use the pretrained feature $\hat{\mathbf{F}}_{DINO}^{\mathcal{I}}$ derived from DINOv2 Oquab et al. (2023), a self-supervised vision foundation model, to address the scale ambiguity Li et al. (2023) between 2D and 3D patches. In the hierarchical architecture, the coarse-level features capture the overall structure of the scene to support broad-scale matching, while the fine-level features provide detailed information for precise matching at a finer level.

**3D Backbone.** We utilize a 3D backbone based on KPConv Thomas et al. (2019) to the point cloud $\mathbf{P}$, producing the cluster-level (coarse level) $\hat{\mathbf{F}}_{3D}^{\mathcal{P}} \in \mathbb{R}^{\hat{N} \times \hat{d}}$ and the point-level (fine level) features $\mathbf{F}_{3D}^{\mathcal{P}} \in \mathbb{R}^{N \times d}$, with the corresponding coordinates represented by $\hat{\mathbf{P}} \in \mathbb{R}^{\hat{N} \times 3}$ and $\mathbf{P} \in \mathbb{R}^{N \times 3}$, respectively.

Additionally, we lift the 2D image into a 3D by applying the monocular depth estimator Zoe-Depth Bhat et al. (2023). Specifically, we first generate a depth map $\mathbf{D}^{\mathcal{I}} \in \mathbb{R}^{H \times W}$ and draw $N^{\mathcal{I}}$ sample points $\mathbf{P}^{\mathcal{I}} = \{\mathbf{p}^{\mathcal{I}}\}$ by

$$\mathbf{p}^{\mathcal{I}} \sim \mathbf{K}^{-1} \cdot \mathbf{D}^{\mathcal{I}} \cdot \mathbf{C}. \tag{2}$$

Then, due to differences in scale between the depth-estimated and original point clouds, these sampled points are processed with a separate encoder. The resulting features are projected back onto the image, generating the patch-level feature $\mathbf{F}_{3D}^{\mathcal{I}} \in \mathbb{R}^{\hat{H} \times \hat{W} \times \hat{d}}$, and the pixel-level feature $\mathbf{F}_{3D}^{\mathcal{I}} \in \mathbb{R}^{H \times W \times d}$., equalizing resolutions for subsequent matching.

**2D-3D Attention.** To bridge the modality gap between image 2D features and point cloud 3D features, we follow the standard cross-attention mechanism introduced in previous work Li et al. (2023). Specifically, the 2D image feature $\hat{\mathbf{F}}_{2D}$ and 3D point cloud feature $\hat{\mathbf{F}}_{3D}^{\mathcal{P}}$ are iteratively processed by applying self-attention and cross-attention. Through this process, we obtain cross-modality features denoted as $\hat{\mathbf{F}}_{2D,Attn}$ and $\hat{\mathbf{F}}_{3D,Attn}^{\mathcal{P}}$.

**Multi-modality Matching Score Mapping.** We compute coarse-level matching scores $\hat{\mathbf{S}} \in \mathbb{R}^{(\hat{H} \times \hat{W}) \times \hat{N}}$ for the 2D-3D and 3D-3D matching. For 2D-3D matching, we compute the matching score $\hat{\mathbf{S}}_{\text{2D-3D}}$ between the 2D image feature $\hat{\mathbf{F}}_{2D,\text{Attn}}$ and the 3D point feature $\hat{\mathbf{F}}_{3D,\text{Attn}}^{\mathcal{P}}$ by

$$\hat{\mathbf{S}}_{\text{2D-3D}} = \hat{\mathbf{F}}_{2D,\text{Attn}} (\hat{\mathbf{F}}_{3D,\text{Attn}}^{\mathcal{P}})^T / \sqrt{\hat{d}}. \tag{3}$$

In the similar way, for 3D-3D matching, we compute the matching score $\hat{\mathbf{S}}_{\text{3D-3D}}$ between the 3D image feature $\hat{\mathbf{F}}_{3D}^{\mathcal{I}}$ and the 3D point feature $\hat{\mathbf{F}}_{3D}^{\mathcal{P}}$ by

$$\hat{\mathbf{S}}_{\text{3D-3D}} = \hat{\mathbf{F}}_{3D}^{\mathcal{I}}(\hat{\mathbf{F}}_{3D}^{\mathcal{P}})^T / \sqrt{\hat{d}}. \tag{4}$$

### 3.4 SCORE AGGREGATION MODULE (SAM)

In this section, we introduce the Score Aggregation Module (SAM), which integrates dual matching scores $\hat{\mathbf{S}}_{\text{2D-3D}}$ and $\hat{\mathbf{S}}_{\text{3D-3D}}$ obtained from MMM into a single unified matching score $\hat{\mathbf{S}}_{\text{fused}}$. This module refines the unified scores by leveraging spatial context, incorporating local correspondence cues from the 2D image while exploiting the inherent spatial relationships of the 3D point cloud.

**Anchor-Pivot 5D Encoder.** Our anchor-pivot 5D encoder takes as input the set of matching score maps $\{\hat{\mathbf{S}}_{\text{2D-3D}}, \hat{\mathbf{S}}_{\text{3D-3D}}\} \in \mathbb{R}^{2 \times (\hat{H} \times \hat{W}) \times \hat{N}}$ and merges them into a single fused matching score, $\hat{\mathbf{S}}_{\text{fused}} \in \mathbb{R}^{(\hat{H} \times \hat{W}) \times \hat{N}}$. To fully leverage the spatial relationships in both 2D and 3D modalities, we accomplish this by building 5D correlation blocks.

However, unlike fixed image coordinates, the spatial coordinates of the 3D point cloud are not static, making it challenging to use a fixed-form kernel for 5D convolution. Furthermore, the 5D correlation network demands significant computational resources due to its high dimensional complexity, as detailed in the Appendix. To address this challenge with an effective and feasible solution, we introduce an anchor-pivot 5D encoder, inspired by the structure of the center-pivot 4D convolution Min et al. (2021).

Our anchor-pivot 5D encoder separates the 2D and 3D kernels, effectively eliminating ambiguities in connections between the 2D and 3D dimensions. This structure enables explicit modeling of the matching relationships among neighboring pixels and points, ensuring that the aggregated correspondences exhibit strong geometric consistency. A detailed architecture of this encoder is depicted in Figure 3.

Given coarse-level coordinates $\hat{\mathbf{C}} = [\hat{\mathbf{c}}]$ and $\hat{\mathbf{P}} = [\hat{\mathbf{p}}]$, where $\hat{\mathbf{c}} \in \mathbb{R}^2$ and $\hat{\mathbf{p}} \in \mathbb{R}^3$ are the elements of $\hat{\mathbf{C}}$ and $\hat{\mathbf{P}}$, respectively, the anchor-pivot 5D encoder block can be formulated by

$$\mathbf{AP}_{\text{5D}}(\hat{\mathbf{S}}(\hat{\mathbf{c}}, \hat{\mathbf{p}})) = \mathcal{E}_{2D}(\hat{\mathbf{S}}(\hat{\mathbf{C}}, \hat{\mathbf{p}})) + \mathcal{E}_{3D}(\hat{\mathbf{S}}(\hat{\mathbf{c}}, \hat{\mathbf{P}}), \hat{\mathbf{P}}), \tag{5}$$

where $\mathcal{E}_{2D}(\cdot)$ and $\mathcal{E}_{3D}(\cdot)$ are the encoder of 2D and 3D, respectively, and their detailed architectures are described in the Appendix.

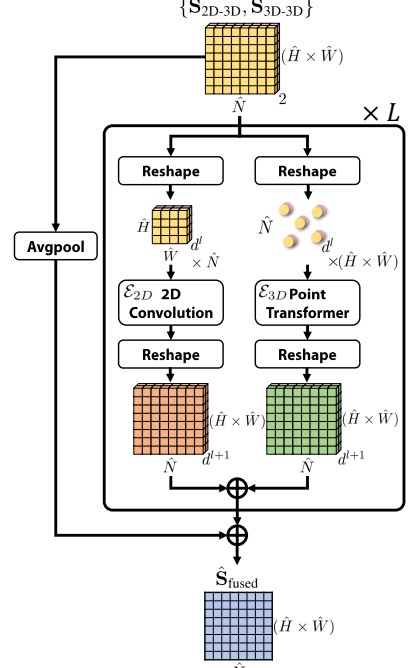

Figure 3: The detailed structure of the anchor-pivot 5D encoder. By splitting the high-dimensional (5D) computation into a 2D convolution for the image side and a point transformer for the point cloud side, this design not only reduces the computational burden but also captures the matching relationships among neighboring pixels and points. Consequently, the encoder promotes geometric consistency in the aggregated matching scores.

When the score map $\mathbf{S}^l \in \mathbb{R}^{d^l \times (\hat{H} \times \hat{W}) \times \hat{N}}$ enters the block, it is reshaped into two separate forms. One is reshaped to $\hat{N} \times d^l \times \hat{H} \times \hat{W}$ to serve as input for the 2D encoder, and the other is reshaped to $(\hat{H} \times \hat{W}) \times d^l \times \hat{N}$ for the 3D encoder. After separately processing these features, both outputs are reshaped back to the original map size, and an element-wise sum is applied to generate the score map $\mathbf{S}^{l+1}$.

By repeatedly processing matching scores through multiple Anchor-Pivot 5D encoder blocks, we progressively incorporate broader spatial context from both 2D and 3D modalities, resulting in an aggregated matching score map $\hat{\mathbf{S}}_{\text{fused}}$ with enhanced structural consistency. Then, we adopt Sinkhorn iterations Cuturi (2013) to compute a soft assignment matrix.

**Coarse and fine correspondence matching.** The coarse-to-fine matching procedure involves two steps. First, we identify coarse correspondences by selecting the top-K matches based on aggregated multi-modal matching scores. Then, within each coarse correspondence region, we perform fine-level matching exclusively within the localized coarse region. Specifically, fine-level pixel-to-point correspondences are estimated by computing cosine similarity between pixel-wise and point-wise feature descriptors. For the pixel-wise representation, we concatenate $\mathbf{F}_{2D}$ and $\mathbf{F}_{3D}^{\mathcal{I}}$, while the point-wise representation is obtained by duplicating $\mathbf{F}_{3D}^{\mathcal{P}}$. Among these fine-level matches, mutual top-K correspondences within each coarse region are selected as the final correspondence set.

## 3.5 Training Objective

We utilize two types of loss functions commonly used in matching tasks: circle loss Sun et al. (2020); Li et al. (2023); Wu et al. (2024); Qin et al. (2022), a type of contrastive loss and focal loss Wu et al. (2024). In the circle loss, for the coarse level, we apply a scaled circle loss Li et al. (2023); Qin et al. (2022) to adaptively adjust the loss based on the degree of overlap between the image and point cloud. To train the similarity of features across various dimensions, 2D-3D circle loss $\mathcal{L}_{coarse}^{2D-3D}$, and 3D-3D circle loss $\mathcal{L}_{coarse}^{3D-3D}$ is used. For the fine level, we use the standard circle loss Li et al. (2023) to achieve precise feature matching. So, $\mathcal{L}_{fine}$ is used to compare the 2D and 3D features of the image with the 3D features of the point cloud. Then, our entire circle loss is $\mathcal{L}_{circle} = \lambda_{coarse}(\mathcal{L}_{coarse}^{2D-3D} + \mathcal{L}_{coarse}^{3D-3D}) + \lambda_{fine}\mathcal{L}_{fine}$. We also adopt focal loss Wu et al. (2024) $\mathcal{L}_{focal}$ for the coarse level by comparing the ground truth of coarse matching relations with our aggregated matching score. Therefore, Our total loss is computed as a weighted sum of the two components: $\mathcal{L}_{total} = \lambda_c\mathcal{L}_{circle} + \lambda_f\mathcal{L}_{focal}$.

# 4 Experiments

## 4.1 Implementation Details

**Backbone.** For the 2D backbone, we use a 4-stage ResNet He et al. (2016) with FPN, where each stage outputs {128, 128, 256, 512} channels. Following Wu et al. (2024), we crop the input image resolution to (476, 630) for compatibility with the DINOv2 network. Then, the patch size at the coarse level is downsampled to (34, 43). For the 3D backbone, we use two 3-stage KPConv Thomas et al. (2019) with each stage outputting {128, 256, 512} channels. The point clouds are initially divided into voxels with a size of 2.5 cm, and the voxel size is doubled progressively at each subsequent stage. Each transformer layer consists of 256 feature channels, utilizes 4 attention heads, and applies ReLU as the activation function. The DINO features are combined with the coarsest-level feature from the ResNet and are also utilized as inputs to the transformer for image feature processing. At the fine level, we utilize the 128-dimensional finest level features from both the 2D encoder and the 3D encoder. By combining these multi-modality features, we perform feature matching in a 256-dimensional space.

**Anchor-Pivot 5D Encoder.** Our anchor-pivot 5D encoder consists of a 4-stage 5D correlation block, with output channels set to {4, 8, 16, 1} for each stage. The $\mathcal{E}_{2D}(\cdot)$ operation employs a ResNet He et al. (2016) structure, while $\mathcal{E}_{3D}(\cdot)$ adopts the Point Transformer Zhao et al. (2021) structure.

**Training detail.** We use the Adam optimizer with a learning rate of $1 \times 10^{-4}$, weight decay of $1 \times 10^{-6}$, and a step learning rate scheduler which decreases the learning rate to 95% every one steps. The network is trained for 20 epochs with batch size 1. We set $\lambda_{coarse} = 1.0$, $\lambda_{fine} = 1.0$, $\lambda_c = 1.0$ and $\lambda_f = 1.0$.

**Dataset.** We evaluate our method on three datasets: RGB-D Scenes V2 Lai et al. (2014), 7-Scenes Glocker et al. (2013), and KITTI-DC Uhrig et al. (2017). The RGB-D Scenes V2 dataset contains indoor image-to-point-cloud pairs with at least 30% overlap, split into 1,748 training, 236 validation, and 497 testing pairs. The 7-Scenes dataset comprises indoor scenes with a minimum 50% overlap, resulting in 4,048 training, 1,011 validation, and 2,304 testing pairs. The KITTI-DC dataset presents outdoor scenarios with sparse LiDAR point clouds, and we created 2,985 training pairs specifically for short-range outdoor registration evaluation. More detailed information is provided in the Appendix.

Table 1: Evaluation results on RGB-D Scenes V2 and 7Scenes. The best scores are highlighted in **boldfaced**, while the second-best are underlined.

| Model | RGB-D Scenes V2 | | | | | 7Scenes | | | | | | | | Mean |
|---|---|---|---|---|---|---|---|---|---|---|---|---|---|---|
| | Scene-11 | Scene-12 | Scene-13 | Scene-14 | Mean | Chess | Fire | Heads | Office | Pupk | Kitc | Stairs | Mean | |
| *Inlier Ratio(%) ↑* | | | | | | | | | | | | | | |
| FCGF-2D3D | 6.8 | 8.5 | 11.8 | 5.4 | 8.1 | 34.2 | 32.8 | 14.8 | 26.0 | 23.3 | 22.5 | 6.0 | 22.8 | 15.5 |
| P2-Net | 9.7 | 12.8 | 17.0 | 9.3 | 12.2 | 55.2 | 46.7 | 13.0 | 36.2 | 32.0 | 32.8 | 5.8 | 31.7 | 22.0 |
| Predator-2D3D | 17.7 | 19.4 | 17.2 | 8.4 | 15.7 | 34.7 | 33.8 | 16.6 | 25.9 | 23.1 | 22.2 | 7.5 | 23.4 | 20.0 |
| 2D3D-MATR | 32.8 | 34.4 | 39.2 | 23.3 | 32.4 | 72.1 | 66.0 | 31.3 | 60.7 | 50.2 | 52.5 | 18.1 | 50.1 | 41.3 |
| FreeReg | 36.6 | 34.5 | 34.2 | 18.2 | 30.9 | - | - | - | - | - | - | - | - | - |
| Diff-Reg | 47.2 | 48.7 | 32.9 | 22.4 | 37.8 | 78.2 | 68.8 | 49.1 | 65.6 | 46.4 | 54.6 | 21.2 | 54.9 | 46.4 |
| CA-I2P | 38.6 | 40.6 | 38.9 | 24.0 | 35.5 | 73.6 | 66.4 | 34.5 | 62.4 | 52.1 | 52.8 | 19.1 | 51.6 | 43.6 |
| Diff²I2P | - | - | - | - | 36.9 | 74.1 | 68.8 | 39.2 | 65.6 | 52.1 | | | 53.2 | 45.1 |
| DuMA(Ours) | **58.2** | **61.4** | **52.0** | **31.1** | **50.7** | **81.1** | **70.0** | **53.6** | **67.6** | 51.9 | **58.5** | 19.5 | **57.5** | **54.1** |
| *Feature Matching Recall(%) ↑* | | | | | | | | | | | | | | |
| FCGF-2D3D | 11.0 | 30.4 | 51.5 | 15.5 | 27.1 | 99.7 | 98.2 | 69.9 | 97.1 | 83.0 | 87.7 | 16.2 | 78.8 | 53.0 |
| P2-Net | 48.6 | 65.7 | 82.5 | 41.6 | 59.6 | 100.0 | 99.3 | 58.9 | 99.1 | 87.2 | 92.2 | 16.1 | 79.0 | 69.3 |
| Predator-2D3D | 44.4 | 41.2 | 21.6 | 13.7 | 30.2 | 91.3 | 95.1 | 76.7 | 88.6 | 79.2 | 80.6 | 31.1 | 77.5 | 53.9 |
| 2D3D-MATR | 98.6 | 98.0 | 88.7 | 77.9 | 90.8 | 100.0 | 99.6 | 98.6 | 100.0 | 92.4 | 95.9 | 58.1 | 92.1 | 91.5 |
| FreeReg | 91.9 | 93.4 | 93.1 | 49.6 | 82.0 | - | - | - | - | - | - | - | - | - |
| Diff-Reg | 100.0 | 100.0 | 88.7 | 77.0 | 91.4 | 100.0 | 100.0 | 98.6 | 100.0 | 90.3 | 98.2 | **64.9** | 93.1 | 92.3 |
| CA-I2P | 100.0 | 100.0 | 91.8 | 82.7 | 93.6 | 100.0 | 100.0 | 98.6 | 100.0 | 92.0 | 95.5 | 60.8 | 92.4 | 93.0 |
| Diff²I2P | - | - | - | - | 77.1 | 100.0 | 100.0 | 100.0 | 100.0 | 93.4 | 96.2 | 55.4 | 92.2 | 84.7 |
| DuMA(Ours) | 100.0 | 100.0 | **100.0** | **84.1** | **96.0** | 100.0 | 100.0 | 100.0 | 100.0 | 90.3 | **99.9** | 58.1 | **93.8** | **94.9** |
| *Registration Recall(%) ↑* | | | | | | | | | | | | | | |
| FCGF-2D3D | 26.4 | 41.2 | 37.1 | 16.8 | 30.4 | 89.5 | 79.7 | 19.2 | 85.9 | 69.4 | 79.0 | 6.8 | 61.4 | 45.9 |
| P2-Net | 40.3 | 40.2 | 41.2 | 31.9 | 38.4 | 96.9 | 86.5 | 20.5 | 91.7 | 75.3 | 82.0 | 4.1 | 65.7 | 52.1 |
| Predator-2D3D | 44.4 | 41.2 | 21.6 | 13.7 | 30.2 | 69.6 | 60.7 | 17.8 | 62.9 | 56.2 | 62.6 | 9.5 | 48.5 | 39.4 |
| 2D3D-MATR | 63.9 | 53.9 | 58.8 | 49.1 | 56.4 | 96.9 | 90.7 | 52.1 | 95.5 | 80.9 | 86.1 | 28.4 | 75.8 | 66.1 |
| FreeReg | 74.2 | 72.5 | 54.5 | 27.9 | 57.3 | - | - | - | - | - | - | - | - | - |
| Diff-Reg | 98.6 | **99.0** | 86.6 | 63.7 | 87.0 | 97.9 | 86.5 | 84.9 | 97.3 | 76.7 | 91.9 | 21.6 | 79.6 | 83.3 |
| CA-I2P | 68.1 | 73.5 | 63.9 | 47.8 | 63.3 | **99.0** | 90.7 | 68.5 | 96.2 | 83.0 | 88.1 | 31.1 | 79.5 | 71.4 |
| Diff²I2P | - | - | - | - | 60.5 | **99.0** | **95.6** | 74.0 | **98.9** | **86.8** | 90.2 | **36.5** | 83.0 | 71.8 |
| DuMA(Ours) | **100.0** | 98.0 | **92.8** | **79.6** | **92.6** | 98.6 | 92.3 | **89.0** | 98.4 | 78.8 | **93.4** | 31.1 | **83.1** | **87.9** |

**Evaluation Metrics.** We use three evaluation metrics to assess the accuracy of image-to-point cloud registration across both indoor and outdoor datasets. (1) **Inlier Ratio (IR)** measures the ratio of pixel-to-point matches with a 3D distance below a specified threshold among all candidate matches. We set this threshold to 5 cm for indoor datasets (e.g., RGB-D Scenes V2, 7Scenes) and 3 m for outdoor datasets (e.g., KITTI-DC). (2) **Feature Matching Recall (FMR)** evaluates the ratio of I2P pairs with an inlier ratio that surpasses a specified threshold (e.g., 10%), indicating the proportion of pairs with sufficiently accurate correspondences. (3) **Registration Recall (RR)** measures the percentage of correctly aligned I2P pairs. We define alignment as RMSE below 10 cm for indoor datasets (e.g., RGB-D Scenes V2, 7Scenes) and translation error under 3 m for KITTI-DC.

## 4.2 EVALUATIONS ON RGB-D SCENES V2

**Comparisions to the state-of-the-arts.** We provide the evaluation results on RGB-D Scenes V2 in Table 1. The results demonstrate that our proposed method, DuMA, achieves the best performance across all three metrics. DuMA achieves a mean score of 50.7% for the Inlier Ratio (IR), which is 12.9% higher than Diff-Reg at 37.8%. Notably, DuMA demonstrates strong performances in Scene-13 and Scene-14, particularly challenging scenarios requiring detailed feature matching. This indicates DuMA's capability in accurately identifying and maintaining correspondences under demanding conditions. In Feature Matching Recall (FMR), DuMA achieves the top score across all scenes, with an impressive average of 96.0%. This high recall rate demonstrates DuMA's effectiveness to find reliable matches across diverse and complex environments. Furthermore, for Registration Recall (RR), DuMA secures a top score of 92.6%, the highest among all tested models, showing its ability to identify precise correspondences required for accurate alignment across varying depth ranges. Notably, DuMA shows remarkable performance improvements in challenging scenes such as Scene-14, highlighting its robust capacity for multi-modal alignment, which is a crucial factor in registration tasks. These impressive results can be attributed to our approach, which explicitly models inter-modal feature relationships through a dedicated score aggregation network. Unlike FreeReg, which processes multi-modal features in parallel without interaction, our method fuses 2D and 3D features to enhance spatial correspondence, resulting in improved alignment accuracy and robustness across diverse scene conditions.

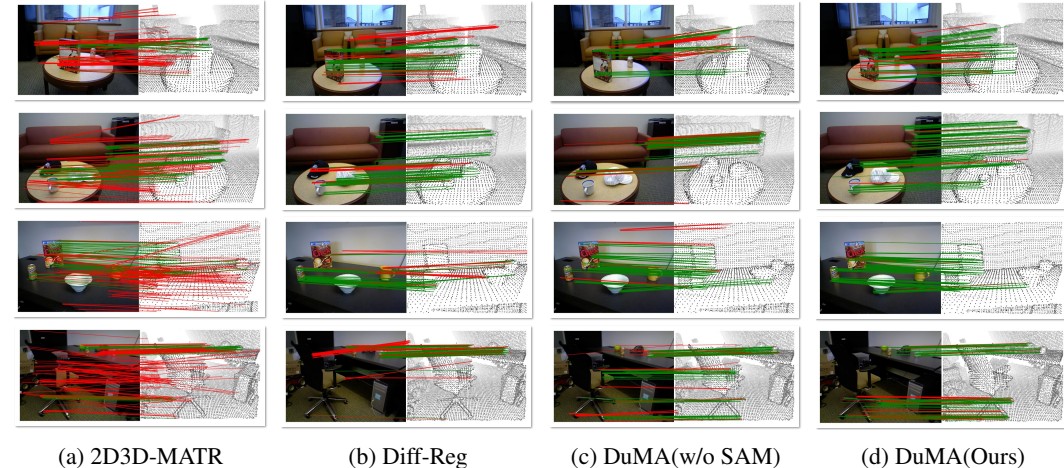

| (a) 2D3D-MATR | (b) Diff-Reg | (c) DuMA(w/o SAM) | (d) DuMA(Ours) |

Figure 4: Qualitative results on RGB-D V2 dataset. Correct / incorrect matches are colored with green / red.

**Qualitative results.** In Figure 4, we compare our approach with the two latest methods and also with the average of our model's dual matching scores. DuMA consistently maintains high matching accuracy even in complex scenes, particularly in environments where objects and backgrounds are intricately intertwined. In contrast, the other two methods exhibit more incorrect matches in complex scenes compared to DuMA. 2D3D-MATR frequently produces incorrect matches, reducing its accuracy in complex scenes. While Diff-Reg achieves relatively high accuracy, its diffusion-based approach to refining the matching matrix often leads to a concentration on specific points. This limitation makes the model overlook broader context, reducing performance in complex scenes.

When using the simple average of our model's dual matching scores, matching tends to occur only in specific areas where both 2D and 3D features are distinctly prominent. Therefore, by merging the two matching scores while incorporating surrounding spatial context, our approach yields more precise correspondences that exhibit enhanced geometric consistency.

### 4.3 EVALUATIONS ON 7SCENES

**Comparisions to the state-of-the-arts.** The evaluation of 7Scenes is shown in Table 1. Overall, DuMA outperforms all compared methods and achieves the best overall results. Additionally, while performance varies across scenes, DuMA consistently achieves strong results by effectively integrating both 2D–3D and 3D–3D matching cues, demonstrating robustness in both complex and sparse feature scenarios.

### 4.4 EVALUATIONS ON KITTI-DC

**Comparisions to the state-of-the-arts.** DuMA outperforms existing methods and is shown to be effective in outdoor environments, with a notable improvement in registration recall as shown in Table 2. This highlights the robustness of our method in handling sparse LiDAR data, enabling more reliable feature matching and registration in challenging outdoor scenarios.

### 4.5 ABLATION STUDIES

In this ablation study on the RGB-D Scenes V2 dataset, we provide a qualitative assessment of the geometric consistency achieved through feature matching score visualization. We also analyze the impact of the fusion weight between 2D and 3D features, demonstrating how different weighting strategies affect the balance between geometric and appearance cues. In the Appendix, we further report ablation studies on (i) the effectiveness of the Multi-Modality Matching and Score Aggregation modules, (ii) the impact of the number of sampling points, (iii) the effect of backbone quality and depth estimation, (iv) different 3D–3D transformation estimation methods, (v) runtime and memory, (vi) the complexity analysis of the anchor–pivot 5D encoder, and (vii) generalization tests.

Table 2: Evaluation results on KITTI-DC. The best scores are highlighted in **boldfaced**, while the second-best are underlined.

| Model | IR(%) | FMR(%) | RR(%) |
|---|---|---|---|
| 2D3D-MATR | 59.1 | 99.7 | 75.4 |
| FreeReg | 58.3 | 99.7 | 70.5 |
| Diff²I2P | 62.9 | 99.7 | 82.2 |
| DuMA(Ours) | **65.8** | **100.0** | **85.9** |

Table 3: Ablation on fusion weight between 2D and 3D features. The best scores are highlighted in **boldfaced**, while the second-best scores are underlined.

| $\alpha$ | 0.0 | 0.2 | 0.4 | 0.5 | 0.6 | 0.8 | 1.0 |
|---|---|---|---|---|---|---|---|
| **IR(%)** | 45.9 | 50.1 | **50.7** | **50.7** | **50.7** | 47.7 | 37.4 |
| **FMR(%)** | **96.0** | **96.0** | **96.0** | **96.0** | **96.0** | **96.0** | 88.7 |
| **RR(%)** | 91.5 | 92.0 | **92.6** | **92.6** | **92.6** | 90.9 | 79.2 |

**Feature Matching Score Visualization.** We visualize the matching scores to assess how our anchor-pivot 5D encoder enhances geometric consistency in the final matching results. To this end, a point cluster from the point cloud is selected as the query, and we visualize the corresponding matching scores in the image, reflecting the contributions from 2D-3D matching, 3D-3D matching, and our anchor-pivot 5D encoder. As shown in Figure 5, when the query cluster is located in areas of the image that are difficult to distinguish from the background, using only 2D-3D matching results in a wide distribution of high matching scores

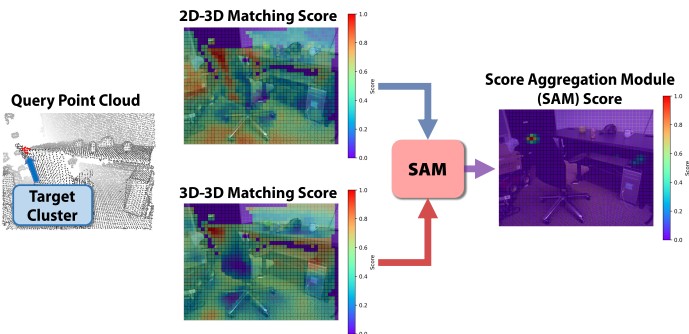

Figure 5: Feature matching score visualization.

across regions with similar colors and features. Conversely, relying solely on 3D-3D matching exploits geometric cues (e.g., edges), concentrating high scores on edge-related areas. Notably, our Anchor-Pivot 5D encoder combines these two perspectives while also considering surrounding spatial information, thereby enhancing geometric consistency in the final score distribution. By merging the complementary information from 2D images and 3D point clouds, the encoder produces matching regions that are both precise and context-aware, yielding robust correspondences even in visually or geometrically challenging scenarios.

**Impact of Feature Fusion Weight** To analyze the contribution of each modality, we conducted a weighted feature fusion experiment in the final block of the Anchor-Pivot 5D encoder, where the fused feature is computed as $\mathbf{f}^{(L)} = \alpha \cdot \mathbf{f}_{2D}^{(L)} + (1-\alpha) \cdot \mathbf{f}_{3D}^{(L)}$. As shown in Table 3, the model performs best when the 2D and 3D features are balanced. When over-relying on one modality (especially 2D) led to a decrease in overall performance. This confirms that jointly leveraging both modalities is crucial for achieving robust registration.

## 5 CONCLUSION

In this paper, we presented DuMA, a novel learnable framework for image-to-point cloud registration that utilizes the complementary strengths of simultaneous 2D-3D and 3D-3D matching. By integrating geometric verification into our score aggregation module, DuMA effectively filters out ambiguous correspondences and preserves structural consistency across modalities. Moreover, our innovative Anchor-Pivot 5D encoder decomposes high-dimensional matching scores into distinct 2D and 3D components, enabling feasible aggregation with reduced computational overhead. Experimental results show that DuMA significantly improves alignment accuracy and robustness, especially in complex environments. Our method still has limitations, as it is sensitive to the quality of depth estimation and struggles in extreme scenarios such as textureless regions where both visual and geometric cues are insufficient. Future work could incorporate depth uncertainty modeling or refinement to further improve robustness, and addressing textureless cases may require integrating additional modalities or stronger priors to resolve the inherent ambiguity.

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

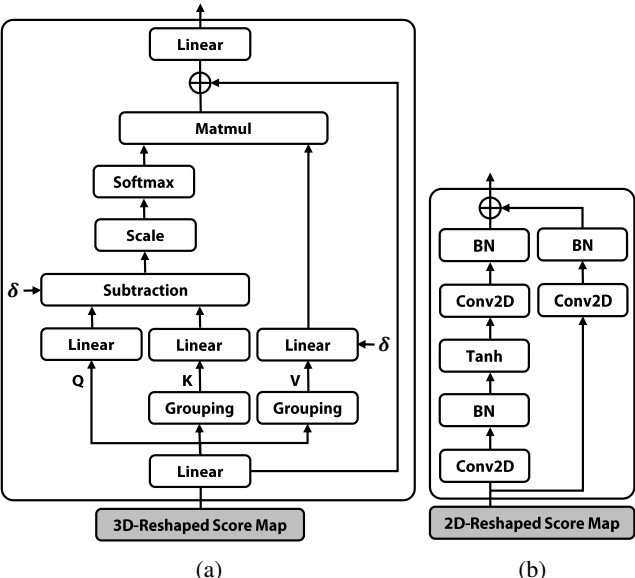

Figure 6: More architectural details of Anchor-Pivot 5D Encoder: (a) Point transformer block Zhao et al. (2021). Grouping: K Nearest Neighbor. $\delta$: Position Encoding. (b) Resnet block He et al. (2016). BN: Batch Normalization Ioffe (2015).

# A APPENDIX

## A.1 ADDITIONAL IMPLEMENTATION DETAILS

**Backbone.** For the 2D backbone, we use a 4-stage ResNet He et al. (2016) with FPN, where each stage outputs $\{128, 128, 256, 512\}$ channels. Following Wu et al. (2024), we crop the input image resolution to (476, 630) to ensure compatibility with the DINOv2 network. Subsequently, the patch size at the coarse level is reduced to (34, 45). Additionally, the coarsest-level feature from the ResNet is combined with the DINO features, which are then passed through a progressive upsampling process to generate pixel-level features. For the 3D backbone, we use two 3-stage KPConv Thomas et al. (2019) with each stage outputting $\{128, 256, 512\}$ channels. The point clouds are initially divided into voxels with a size of 2.5 cm, and the voxel size is doubled progressively at each subsequent stage. For the 3D backbone input from the image, the number of sampling points $N^{\mathcal{I}}$ through the depth map is set to 30,000.

**Anchor-Pivot 5D Encoder.** Our anchor-pivot 5D encoder comprises a 4-stage 5D correlation block, where the output channels for each stage are set to $\{4, 8, 16, 1\}$. The $\mathcal{E}_{3D}(\cdot)$ use the Point Transformer Zhao et al. (2021) structure, while $\mathcal{E}_{2D}(\cdot)$ adopts a ResNet He et al. (2016) structure. We choose ResNet for its proven effectiveness in structured, grid-based image feature extraction, and Point Transformer for its inherent ability to handle irregular, unordered point cloud data through self-attention mechanisms. This encoder combination naturally suits the distinct characteristics of each modality, facilitating effective geometric verification at the coarse matching stage. Our detailed architecture of the anchor-pivot 5D encoder is illustrated in Figure 6. Given the matching score map $\mathbf{S}^l \in \mathbb{R}^{d^l \times (\hat{H} \times \hat{W}) \times \hat{N}}$, it is transformed into two distinct shapes within the block. The first shape is $\hat{N} \times d^l \times \hat{H} \times \hat{W}$, which is fed into the 2D encoder, while the second is reshaped to $(\hat{H} \times \hat{W}) \times d^l \times \hat{N}$ for input to the 3D encoder. Then, within each encoder, batch-wise computations are performed on the reshaped score maps. In the 3D encoder, we perform attention based on the K nearest neighbor (K-NN) search, considering the information from surrounding points to generate features. In this process, we set K=3 to capture local dependencies. Finally, we obtains the output score map $\mathbf{S}^{l+1} \in \mathbb{R}^{d^{l+1} \times (\hat{H} \times \hat{W}) \times \hat{N}}$.

### A.2 LOSS FUNCTION

**Circle Loss.** We apply three types of circle loss at both the coarse and fine levels: 2D-3D patch matching loss at the coarse level $\mathcal{L}_{coarse}^{2D-3D}$, 3D-3D patch matching loss at the coarse level $\mathcal{L}_{coarse}^{3D-3D}$, and pixel-to-point matching loss at the fine level $\mathcal{L}_{fine}$.

For a given target descriptor $d_i$, the descriptors of its positive and negative pairs are denoted by $\mathcal{D}_i^{\mathcal{P}}$ and $\mathcal{D}_i^{\mathcal{N}}$, respectively. The general form of the circle loss for $d_i$ is defined as:

$$\mathcal{L}_{\text{circle}} = \frac{1}{N} \sum_{i=1}^{N} \frac{1}{\gamma} \log \left[ 1 + \sum_{d_j \in \mathcal{D}_i^{\mathcal{P}}} e^{\beta_p^j \left( d_i^j - \Delta_p \right)} \cdot \sum_{d_k \in \mathcal{D}_i^{\mathcal{N}}} e^{\beta_n^k \left( \Delta_n - d_i^k \right)} \right] \tag{6}$$

where $d_i^j$ represents the $\ell_2$-norm feature distance between the anchor descriptor $d_i$ and its positive pair $d_j$, and $d_i^k$ is similarly defined but for the negative pairs. The individual weights for positive pairs, $\beta_p^j = \gamma \lambda_p (d_i^j - \Delta_p)$, and for negative pairs, $\beta_n^k = \gamma \lambda_n (\Delta_n - d_i^k)$, where $\lambda_p$ and $\lambda_n$ are scaling factors for positive and negative pairs, respectively. The terms $\Delta_p$ and $\Delta_n$ are margins that control the influence of positive and negative samples.

Following Li et al. (2023), positive and negative samples are identified based on the overlapping ratio. At the coarse level, if the patch overlapping ratio between 2D and 3D patches is at least 30%, it is regarded as positive, while a ratio below 20% is regarded as negative. Additionally, $\lambda_p$ is defined as the overlapping ratio, while $\lambda_n$ is set to 1. At the fine level, a pixel-point pair is regarded as positive if the 3D distance is within 3.75 cm and the 2D distance is within 8 pixels. Conversely, it is identified as negative if the 3D distance exceeds 10 cm or the 2D distance exceeds 12 pixels. At this level, both $\lambda_p$ and $\lambda_n$ are set to 1. Furthermore, the margins are defined as $\Delta_p = 0.1$ and $\Delta_n = 1.4$.

**Focal Loss.** For the matching score map $\hat{\mathbf{S}}_{\text{fused}} \in \mathbb{R}^{(\hat{H} \times \hat{W}) \times \hat{N}}$ obtained after score aggregation at the coarse level, we use the focal loss Wu et al. (2024) $\mathcal{L}_{focal}$.

Specifically, we define focal loss as

$$\mathcal{L}_{\text{focal,p}} = -\alpha \sum_{i \in \mathcal{P}} (1 - \hat{\mathbf{S}}_{\text{fused}}(i))^{\gamma} \cdot \log(\hat{\mathbf{S}}_{\text{fused}}(i)) \tag{7}$$

$$\mathcal{L}_{\text{focal,n}} = -\alpha \sum_{i \in \mathcal{N}} (\hat{\mathbf{S}}_{\text{fused}}(i))^{\gamma} \cdot \log(1 - \hat{\mathbf{S}}_{\text{fused}}(i)) \tag{8}$$

$$\mathcal{L}_{\text{focal}} = w_{\text{pos}} \cdot \frac{\mathcal{L}_{\text{focal,p}}}{|\mathcal{P}|} + w_{\text{neg}} \cdot \frac{\mathcal{L}_{\text{focal,n}}}{|\mathcal{N}|} \tag{9}$$

where $\mathcal{P}$ represents the set of positive locations in the ground truth, while $\mathcal{N}$ denotes the set of negative locations. Here, $\gamma$ is a focusing parameter, with the weights for positive and negative samples represented by $w_{\text{pos}}$ and $w_{\text{neg}}$, respectively. Specifically, $\alpha$ and $\gamma$ are set to 0.25 and 2.0, while both $w_{\text{pos}}$ and $w_{\text{neg}}$ are set to 1.

### A.3 DETAILED DATASETS

**RGB-D Scenes V2.** RGB-D Scenes V2 Lai et al. (2014) consists of 11,427 RGB-D frames captured across 14 indoor scenes. We use the training data preprocessed by Li et al. (2023), where image-to-point-cloud pairs are generated by creating point cloud fragments from every 25 consecutive depth frames and sampling an RGB image at the same interval. Only image-point-cloud pairs with an overlap ratio of at least 30% are retained. The dataset is divided into training, validation, and testing sets based on scene numbers: scenes 1-8 for training, scenes 9-10 for validation, and scenes 11-14 for testing, resulting in 1,748 training pairs, 236 validation pairs, and 497 testing pairs.

**7-Scenes.** 7-Scenes Glocker et al. (2013) consists of 46 RGB-D sequences captured across 7 indoor scenes. We adopt the training data prepared in Li et al. (2023), Image-to-point-cloud pairs are generated by creating point cloud fragments and sampling RGB images at regular intervals, retaining only those pairs with an overlap ratio of at least 50%. The dataset is divided into training,

validation, and testing sets based on the official sequence split, resulting in 4,048 training pairs, 1,011 validation pairs, and 2,304 testing pairs.

**KITTI-DC.** The KITTI-DC dataset Uhrig et al. (2017) consists of 342 image-to-point cloud (I2P) pairs captured across four outdoor driving scenes. Unlike the RGB-D Scenes V2 and 7Scenes datasets, which primarily contain indoor environments with dense point clouds, KITTI-DC presents a more challenging setting with sparse point clouds obtained from a 64-line LiDAR scan. We use the dataset as processed in Wang et al. (2024), where the distance between each I2P pair is less than 10 meters, making it suitable for evaluating short-range outdoor registration performance. For training, we generated a total of 2,985 training pairs from seven distinct scenes.

## A.4 DETAILED METRICS

Given a 3D point cloud $\mathbf{P} \in \mathbb{R}^{N \times 3}$ and a 2D image $\mathbf{I} \in \mathbb{R}^{H \times W \times 3}$, DuMA estimates correspondences $\mathcal{C} = \{(x_i, y_i) \mid x_i \in \mathbb{R}^3, y_i \in \mathbb{R}^2\}$ between 3D points and 2D pixels. Following Li et al. (2023), we evaluate the estimated correspondences based on three metrics.

**Inlier Ratio (IR).** IR represents the ratio of inliers to all putative pixel-point correspondences $(x_i, y_i) \in \mathcal{C}$. A correspondence is considered an inlier if its 3D Euclidean distance is below the threshold $\tau_1 = 5\,\mathrm{cm}$ under the ground-truth transformation $\mathbf{T}^*$:

$$\mathrm{IR} = \frac{1}{|\mathcal{C}|} \sum_{(x_i, y_i) \in \mathcal{C}} 1(\left\| \mathbf{T}^*(\mathbf{x}_i) - \mathcal{K}^{-1}(\mathbf{y}_i) \right\|_2 < \tau_1), \tag{10}$$

where $1()$ is the indicator function, and $\mathcal{K}^{-1}$ is the function that converts a pixel into a 3D point based on its depth value.

**Feature Matching Recall (FMR).** FMR measures the fraction of image-point cloud pairs whose IR exceeds the threshold $\tau_2 = 0.1$:

$$\mathrm{FMR} = \frac{1}{N} \sum_{i=1}^{N} 1(\mathrm{IR}_i > \tau_2), \tag{11}$$

where $N$ is the number of all point-image pairs in the test dataset.

**Registration Recall (RR).** Registration Recall (RR) measures the fraction of correctly aligned image-point cloud pairs based on the putative correspondences. A pair is considered correctly aligned if the Root Mean Square Error (RMSE) between the point clouds after applying the ground-truth transformation and the predicted transformation $\mathbf{T}$ is below the threshold $\tau_3 = 0.1\,\mathrm{m}$:

$$\mathrm{RMSE} = \sqrt{\frac{1}{|\mathcal{P}|} \sum_{\mathbf{p}_i \in \mathcal{P}} \left\| \mathbf{T}(\mathbf{p}_i) - \mathbf{T}^*(\mathbf{p}_i) \right\|_2^2}, \tag{12}$$

$$\mathrm{RR} = \frac{1}{M} \sum_{i=1}^{M} 1(\mathrm{RMSE}_i < \tau_3). \tag{13}$$

## A.5 ADDITIONAL ABLATION STUDIES AND ANALYSIS

### A.5.1 EFFECTIVENESS OF MULTI-MODALITY MATCHING MODULE (MMM) AND SCORE AGGREGATION MODULE (SAM).

The experimental results in Table 4 demonstrate that the integration of our anchor-pivot 5D encoder with the combined matching strategy yields the highest overall performance. Interestingly, when an anchor-pivot 5D encoder is applied to a single modality, the IR decreases slightly. This is because the encoder introduces a more rigorous geometric verification process that filters out some potential correspondences, although the remaining matches are more reliable, leading to an increase in RR and overall alignment performance.

Table 4: Ablation on effectiveness of MMM and SAM. The best scores are highlighted in **boldfaced**, while the second-best are underlined.

| Method | IR(%) | FMR(%) | RR(%) |
|---|---|---|---|
| 2D-3D | 46.3 | 94.2 | 89.3 |
| 2D-3D + AP$_{5D}$ | 42.2 | 95.0 | 90.3 |
| 3D-3D | 36.2 | 94.0 | 84.5 |
| 3D-3D + AP$_{5D}$ | 32.6 | 92.3 | 86.1 |
| 2D-3D / 3D-3D | 49.5 | 94.3 | 89.0 |
| 2D-3D / 3D-3D + AP$_{5D}$ | **50.7** | **96.0** | **92.6** |

Table 5: Ablation on effectiveness of architectural designs in SAM. The best scores are highlighted in **boldfaced**, while the second-best are underlined.

| Method | IR(%) | FMR(%) | RR(%) |
|---|---|---|---|
| Average | 49.5 | 94.3 | 89.0 |
| MLP | 50.5 | 95.7 | 90.8 |
| Only 2d Encoder | 49.2 | 95.7 | 89.2 |
| Only 3d Encoder | 48.8 | 94.6 | 90.4 |
| Late Fusion | 48.1 | **96.1** | 89.6 |
| Shared Attention | 49.6 | 94.8 | 89.8 |
| AP$_{5D}$ | **50.7** | 96.0 | **92.6** |

### A.5.2 EFFECTIVENESS OF ARCHITECTURAL DESIGNS IN SCORE AGGREGATION MODULE (SAM).

We conduct an ablation study to evaluate various fusion strategies for aggregating the dual matching scores, as summarized in Table 5. Using a simple averaging baseline, we observe moderate performance across all metrics. Replacing this with a learnable MLP-based fusion improves both the inlier ratio (IR) and registration recall (RR), suggesting that non-linear integration of dual cues provides better correspondence estimation.

To further investigate fusion strategies, we implement four additional baselines. 'Only 2D Encoder' and 'Only 3D Encoder' use a single modality by removing the other branch from the pipeline. These models show inferior performance, highlighting the importance of multi-modal interaction for accurate matching. The 'Late Fusion' strategy employs independent 2D and 3D encoders and combines matching scores only at the final stage, without intermediate interaction. Although it achieves the highest feature matching recall (FMR), it performs worse in IR and RR due to the lack of joint spatial reasoning during encoding. The 'Shared Attention' method adopts a cross-modal attention mechanism inspired by Hertz et al. (2024), where both modalities attend to a common latent representation. While this design enables early interaction between views, it does not explicitly model spatial alignment between modalities, resulting in slightly lower overall performance compared to our method.

Our proposed AP-5D encoder achieves the best performance across all metrics, demonstrating its ability to effectively leverage both geometric and visual cues through spatially decomposed and harmonized aggregation. Notably, it yields the highest registration recall (92.6%), validating its strength in preserving reliable correspondences across complex scene structures.

### A.5.3 IMPACT OF NUMBER OF SAMPLING POINTS

We explore the impact of the number of sampling points $N^{\mathcal{I}}$ projected from the depth map in the image. The results are reported in Table 6. The result shows that IR tends to decrease as the number of sampling points increases, and the performance of Registration Recall (RR) no longer improves beyond a certain number of sampling points. The highest performance for RR occurs at 30k sampling points, which indicates that this is the optimal number of sampling points. These results suggest

Table 6: Ablation on different number of sampling points. The best scores are highlighted in **bold-faced**, while the second-best scores are underlined.

| # | IR(%) | FMR(%) | RR(%) |
|---|-------|--------|-------|
| 10K | **53.6** | **96.3** | 91.6 |
| 20K | 51.8 | 95.7 | 91.1 |
| 30K | 50.7 | 96.0 | **92.6** |
| 40K | 52.3 | 96.1 | 90.0 |
| 50K | 48.3 | 96.1 | 89.9 |
| 60K | 46.7 | 95.2 | 86.7 |
| 70K | 44.3 | 94.6 | 85.3 |
| 80K | 48.9 | 94.9 | 90.3 |

Table 7: Ablation on (a)Backbone quality, (b)Depth estimation. The best scores are highlighted in **boldfaced**, while the second-best scores are underlined.

| Method | IR(%) | FMR(%) | RR(%) |
|--------|-------|--------|-------|
| (a) Pretrained Backbone | 53.1 | 96.8 | 92.6 |
| (b) GT Depth | **60.9** | **97.2** | **94.4** |
| DuMA | 50.7 | 96.0 | 92.6 |

that when too many points are sampled, the overlap between points increases, making it difficult to extract the appropriate geometric features. Therefore, for optimal matching performance, it is more effective to use an appropriate number of sampling points rather than excessively increasing the number of points.

### A.5.4 IMPACT OF BACKBONE QUALITY AND DEPTH ESTIMATION

To evaluate the impact of backbone representations, we conducted experiments using pretrained 2D (ImageNet) and 3D (FCGF on 3DMatch) encoders. As shown in Table 7-(a), while feature matching quality improves slightly, the final registration accuracy remains largely unchanged. This indicates that the robustness of our method primarily arises from the proposed dual-view aggregation framework rather than the pretrained features.

In addition, we performed oracle experiments using ground-truth depth to analyze the influence of depth estimation quality. As shown in Table 7-(b), performance improves under accurate depth, confirming that better depth predictions lead to more reliable correspondences. While our current focus is on the matching framework itself, We leave the incorporation of depth uncertainty modeling or refinement to future work to further improve robustness.

### A.5.5 3D-3D TRANSFORMATION ESTIMATION METHOD

Using Zoe-Depth Bhat et al. (2023), we generate a depth map $\mathbf{D}^{\mathcal{I}}$ from the image, enabling the mapping of pixel-to-point correspondences into 3D point correspondences. By leveraging these 3D point-to-point matches, we compute the SE(3) relative pose using the Kabsch algorithm Kabsch (1976). Leveraging the Kabsch algorithm, the transform can be solved given the estimated correspondences $\mathcal{C} = \{(x_i, y_i) \,|\, x_i \in \mathbb{R}^3, y_i \in \mathbb{R}^2\}$ between 3D points and 2D pixels, as defined by:

$$\min_{R,t} \sum_{(x_i,y_i)\in C} \left\| x_i - RProj^{-1}(y_i, d_{y_i}^{\mathcal{I}}, K) + t \right\|^2, \tag{14}$$

where $Proj^{-1}(y_i, \mathbf{D}^{\mathcal{I}}, K)$ lifts the 2d image pixel to 3d point using the depth $d_{y_i}^{\mathcal{I}}$ and the intrinsic matrix $K$. The results are presented in Table 8. We observe that the performance of the transformation estimated using the Kabsch algorithm is inferior to that of PnP. This result arises due to the scale discrepancy between the depth map predicted by Zoe-Depth and the actual depth values, which prevents the generation of points at identical locations. In other words, it demonstrates that a more

Table 8: Ablation on different transformation estimation methods. The best scores are highlighted in **boldfaced**, while the second-best are underlined.

| Method | Scene-11 | Scene-12 | Scene-13 | Scene-14 | Mean |
|---|---|---|---|---|---|
| Mean depth (m) | 1.74 | 1.66 | 1.18 | 1.39 | 1.49 |
| *Registration Recall(%)* ↑ | | | | | |
| FreeReg +Kabsch | 38.7 | 51.6 | 30.7 | 15.5 | 34.1 |
| FreeReg +PnP | 74.2 | 72.5 | 54.5 | 27.9 | 57.3 |
| DuMA+Kabsch | 62.5 | 83.3 | 49.5 | 29.2 | 56.1 |
| DuMA+PnP | **100.0** | **98.0** | **92.8** | **79.6** | **92.6** |

Table 9: Runtime and memory.

| Method | Time (s)↓ | # of Parameters↓ |
|---|---|---|
| 2D3D-MATR | 0.099 | 31.05M |
| Diff-Reg | 0.564 | 373.60M |
| DuSA-Reg | 0.648 | 35.60M |

Table 10: The number of parameters of each module.

| Layer | # of Parameters |
|---|---|
| 2D Encoder | 17.59M |
| 3D Encoder | 1.49M |
| Transformer | 3.91M |
| DINO+Linear | 11.12M |
| AP-5D | 9.44K |

effective approach is to indirectly use the geometric features of the point cloud generated through depth estimation, rather than directly using the point cloud itself for matching.

### A.5.6  RUNTIME AND MEMORY

We present a comparison of runtime and model size with 2D3D-MATR Li et al. (2023) and Diff-Reg Wu et al. (2024) in Table 9. The runtime is measured on a machine equipped with an Intel Xeon Gold 6226R 2.90GHz CPU and a single Nvidia RTX A5000 GPU, using a batch size of 1. In Table 9, our method shows a slightly longer runtime than Diff-Reg, but requires substantially fewer parameters. This indicates that, while there is a minor increase in runtime, our architecture remains more compact compared to existing approaches.

Regarding memory usage, our model has fewer parameters compared to Diff-Reg, even though it includes the additional encoder and the introduction of the anchor-pivot 5D encoder. Furthermore, Table 10 shows that our anchor-pivot 5D encoder contains significantly fewer parameters than other modules. This suggests that the anchor-pivot 5D encoder can generate high-quality correlation information with a minimal number of parameters.

### A.5.7  COMPLEXITY ANALYSIS OF THE ANCHOR–PIVOT 5D ENCODER

A naïve 5D encoder would require a $K_H \times K_W \times K_N$ kernel to be applied at every $\hat{H} \times \hat{W} \times \hat{N}$ location, incurring $\mathcal{O}(\hat{H}\hat{W}\hat{N}K_HK_WK_NC_{\text{in}}C_{\text{out}})$ operations, which is computationally infeasible since $\hat{N}$ is also very large. In contrast, our Anchor–Pivot design decomposes this into two branches: a $K_H \times K_W$ 2D convolution over $\hat{H} \times \hat{W}$ (image anchors) and a $K_N$ 3D operation over $\hat{N}$ (point anchors). The total complexity becomes $\mathcal{O}(\hat{H}\hat{W}K_HK_WC_{\text{in}}C_{\text{out}} + \hat{N}K_NC_{\text{in}}C_{\text{out}})$. This significantly reduces the computation while preserving spatial interactions across modalities.

Table 11: Ablation on Training Data. The best scores are highlighted in **boldfaced**, while the second-best scores are underlined.

| Method | IR(%) | | FMR(%) | | RR(%) | |
|---|---|---|---|---|---|---|
| | **100%** | **10%** | **100%** | **10%** | **100%** | **10%** |
| 2D3D-MATR | 32.4 | 6.9 | 90.8 | 20.7 | 56.4 | 5.2 |
| Diff-Reg | 37.8 | 13.3 | 91.4 | 63.4 | 87.0 | 40.5 |
| DuMA | **50.7** | **30.4** | **96.0** | **89.4** | **92.6** | **78.6** |

### A.5.8 GENERALIZATION TEST

To evaluate the generalization ability under limited supervision, we trained DuMA and the baselines using only 10% of the training data. As shown in Table 11, our method retains strong performance, while baselines suffer a significant drop. We believe this robustness stems from our architectural design, which jointly captures 2D and 3D cues early, enabling more effective convergence.

### A.6 ADDITIONAL FEATURE MATCHING SCORE VISUALIZATION

We further provide additional visualizations of the feature matching scores in Figure 7. These visualizations complement the main paper by offering more examples of how our method differentiates between regions using 2D-3D matching, 3D-3D matching, and the integrated Anchor-Pivot 5D encoder. In particular, we include extra cases where the query point clusters are located in both ambiguous and distinct regions of the image, demonstrating that our encoder consistently fuses the complementary strengths of 2D-3D and 3D-3D matching to yield refined and geometrically coherent correspondences.

### A.7 ADDITIONAL QUALITATIVE RESULTS

Additional Qualitative Results on RGB-D Scenes V2, 7Scenes, and KITTI-DC are shown in Figure 8, Figure 9, and Figure 10, respectively. In Figure 8, our method demonstrates more accurate and global matching compared to 2D3D-MATR and Diff-Reg. Furthermore, we observe robust matching performance even in cases with significant pose differences. Moreover, in Figure 9, we can see accurate and consistent matching performance on the 7Scenes dataset, which exhibits a larger pose variance. Additionally, in Figure 10, our approach maintains strong performance on the KITTI-DC dataset, effectively handling outdoor environments with dynamic elements and large-scale scene variations.

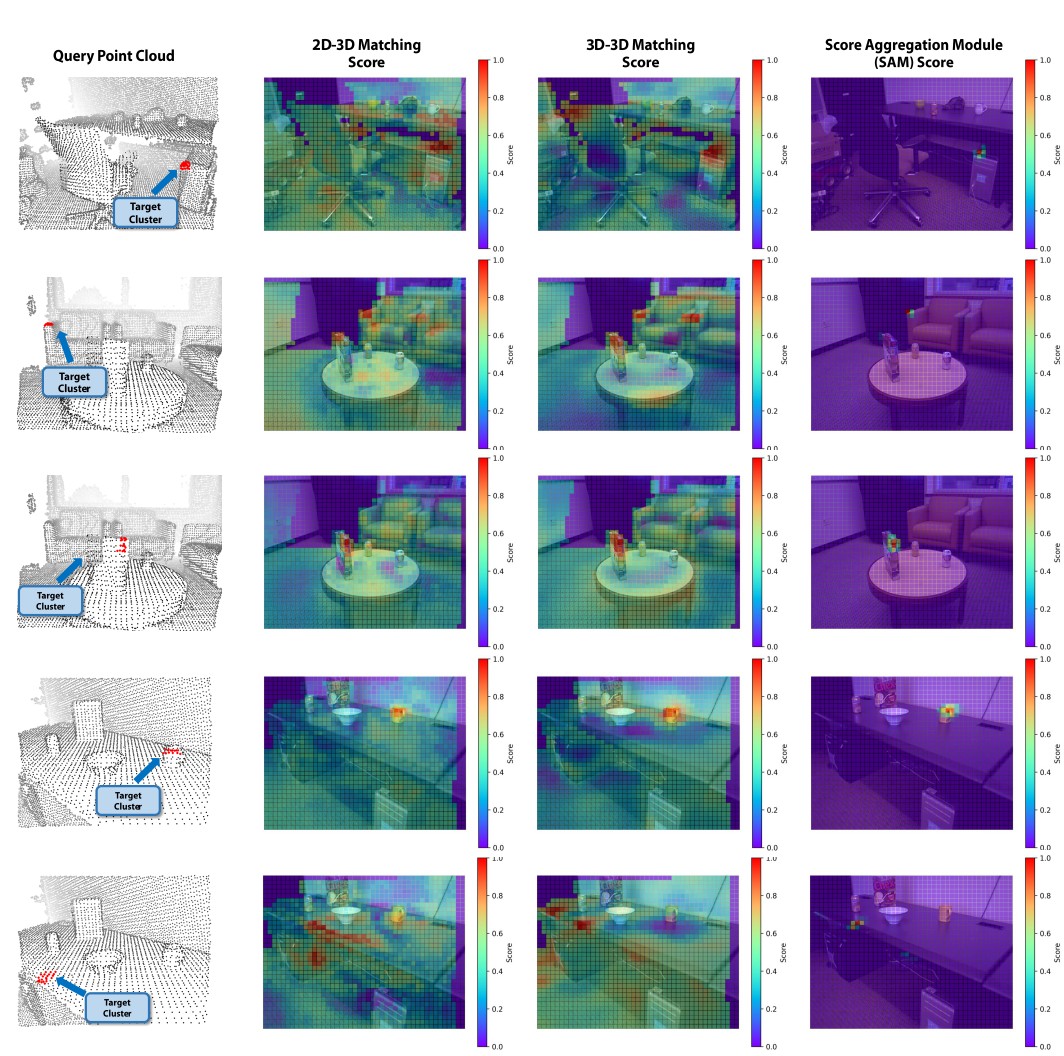

Figure 7: Additional feature matching score visualization

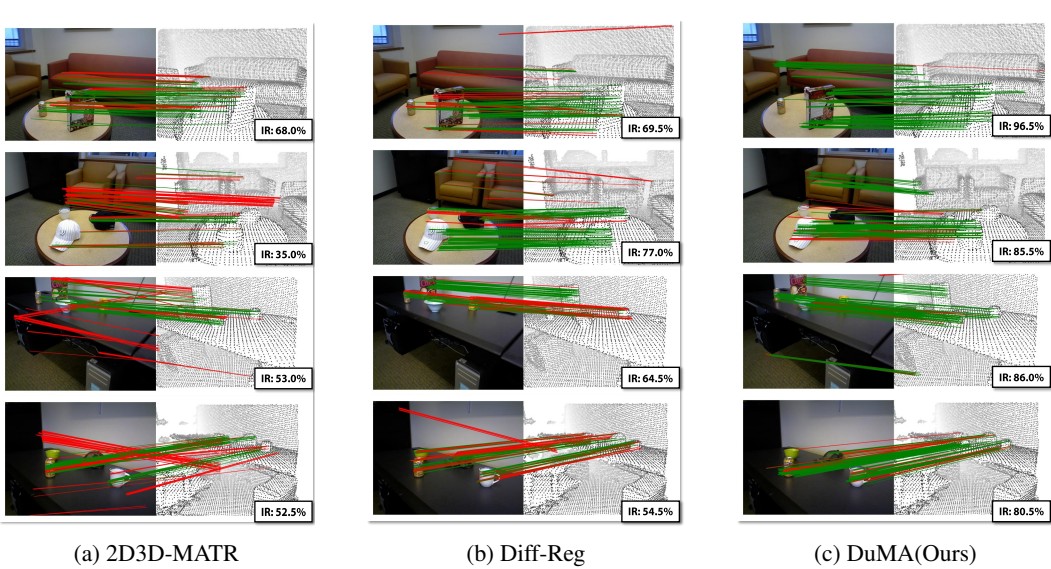

(a) 2D3D-MATR                    (b) Diff-Reg                    (c) DuMA(Ours)

Figure 8: Additional qualitative results on RGB-D V2 dataset. Correct / incorrect matches are colored with green / red.

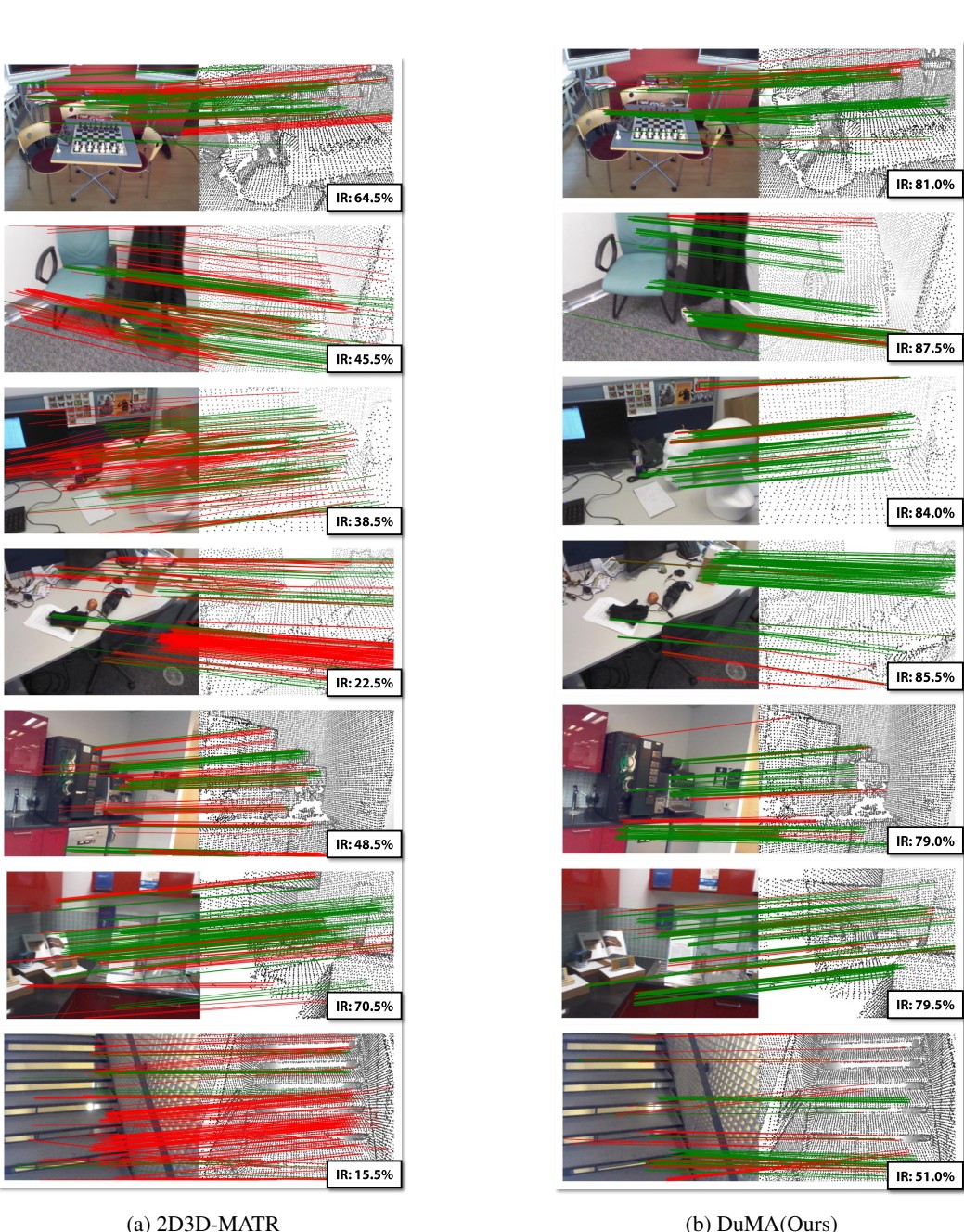

(a) 2D3D-MATR             (b) DuMA(Ours)

Figure 9: Qualitative results on 7Scenes dataset. Correct / incorrect matches are colored with green / red.

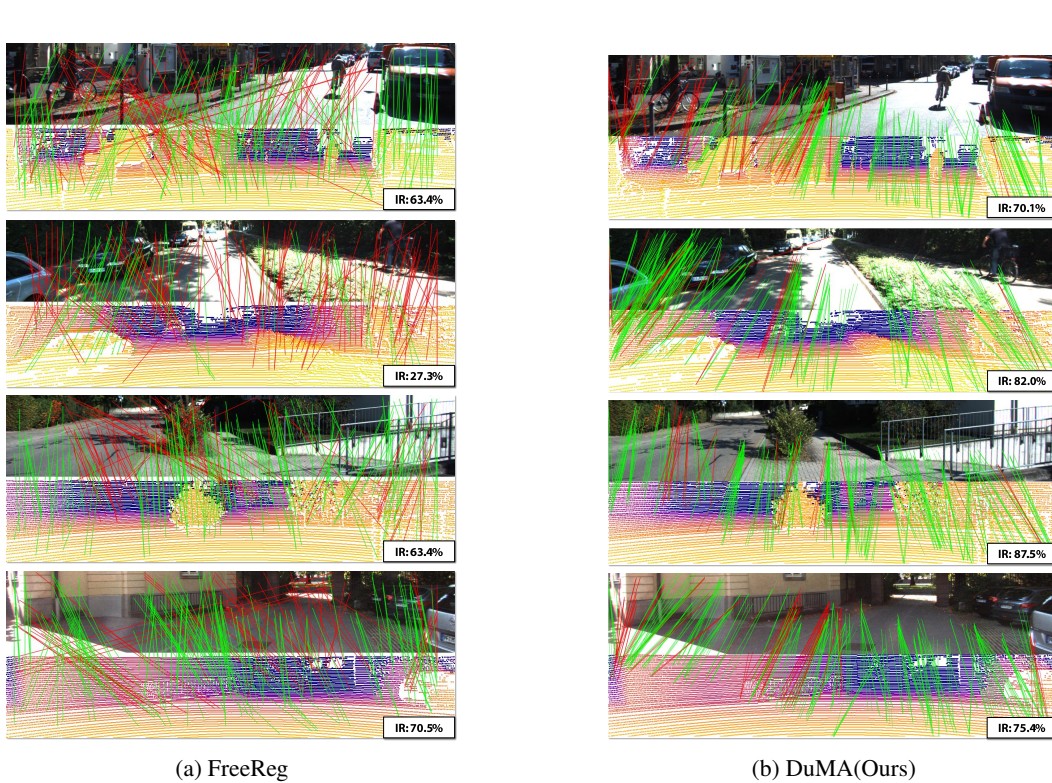

(a) FreeReg                                    (b) DuMA(Ours)

Figure 10: Qualitative results on KITTI-DC dataset. Correct / incorrect matches are colored with green / red.

