# OpenReview forum: "DuMA: Dual Matching Aggregation for Image-to-Point Cloud Registration"
_ICLR.cc/2026/Conference — ICLR 2026 Conference Withdrawn Submission_

### Official Review · Reviewer_KWX6 · 2025-10-30

**Soundness:** 3
**Presentation:** 3
**Contribution:** 2
**Rating:** 4
**Confidence:** 4

**Summary:**

The paper proposes a novel image-to-point cloud registration framework named DuMA (Dual Matching Aggregation), aiming to address ambiguity in cross-modal registration. The core idea is to simultaneously leverage two complementary matching cues: 2D-3D matching: match the visual features of the image (e.g., texture, color) with the geometric features of the point cloud. 3D-3D matching: first use a monocular depth estimator (Zoe-Depth) to lift the 2D image into an “estimated point cloud,” and then perform purely geometric matching between this estimated point cloud and the target point cloud. To fuse the scores of these two matchings (2D-3D and 3D-3D), the authors design a Score Aggregation Module (SAM). Considering that fusing scores from the 2D image space (H×W) and the 3D point cloud space (N) incurs extremely high (5D) computational complexity, the authors further propose an efficient Anchor-Pivot 5D (AP-5D) encoder. This encoder decomposes the 5D aggregation operation into parallel 2D convolutions (on the image grid) and 3D point cloud Transformers (on point clusters), thereby maintaining computational feasibility while capturing cross-modal spatial context.
Experiments on multiple indoor and outdoor datasets such as RGB-D Scenes V2, 7Scenes, and KITTI-DC show that DuMA significantly outperforms existing SOTA methods on key metrics such as registration recall (RR) and inlier ratio (IR).

**Strengths:**

1. Novel “dual matching” aggregation strategy: The biggest contribution of this paper is the strategy of simultaneously using two matching branches—2D-3D (visual–geometric) and 3D-3D (estimated geometry–real geometry). These two cues are highly complementary. As shown in Figure 5, 2D-3D matching is effective in texture-rich regions but produces ambiguity in visually similar areas (such as tabletops); whereas 3D-3D matching focuses more on geometric edges and structures. DuMA combines the two, significantly improving the robustness of matching.
2. Efficient 5D aggregation encoder (AP-5D): How to feasibly fuse matching scores from the 2D image space and the 3D point cloud space is a core challenge. The proposed Anchor-Pivot 5D encoder is a clever solution. It decomposes 5D aggregation into two parallel low-dimensional operations (2D convolution + 3D Transformer), with an elegant structure and high computational efficiency. Appendix Table 10 shows that the AP-5D module itself has only about 9.44K parameters, making it very lightweight.
3. Solid experimental performance: The method achieves SOTA performance on multiple standard benchmarks (RGB-D V2, 7Scenes, KITTI-DC). For example, on RGB-D V2 the registration recall (RR) reaches 92.6%, significantly higher than Diff-Reg (87.0%) and FreeReg (57.3%) (see Table 1), demonstrating the effectiveness of the proposed framework.

**Weaknesses:**

1. Strong dependency on depth estimation quality: The entire 3D-3D matching branch fully relies on the output of a pretrained monocular depth estimator. The authors also indicate that using GT depth would further improve performance (RR from 92.6% to 94.4%), which confirms the method’s sensitivity to depth estimation errors. The paper lacks a quantification of how failure cases (textureless regions, transparent/reflective surfaces, long range, motion blur) affect the 3D-3D scores, as well as an analysis of the stability of SAM under noisy/biased depth.
2. Scale inconsistency in 3D-3D matching: The paper mentions that, due to scale differences between the depth-estimated point cloud and the original point cloud, a separate 3D encoder is required to process the estimated point cloud. The paper could further explore how SAM learns to reconcile these two 3D representations at different scales.
3. Lack of empirical support for comparing depth estimators: The paper does not conduct a systematic comparison across different monocular depth estimators (such as the clearly stronger depth pro moge, etc.), nor does it report registration robustness curves with respect to variations in “depth error/scale bias.”
4. Depth uncertainty: Since GT depth brings significant gains, can SAM assign confidence modeling to the depth-guided 3D-3D scores (e.g., weights or temperature based on scale/noise) to mitigate the impact of Zoe-Depth’s errors?
5. Choice of K: The 3D path uses K=3 by direct setting but without sensitivity or speed-accuracy trade-off analysis. It is recommended to conduct systematic ablations for K=1/3/5/7.
6. The current experimental design does not sufficiently cover challenging scenarios such as low overlap/weak co-visibility. Specifically, the data splits set relatively high overlap thresholds (RGB-D ≥30%, 7-Scenes ≥50%), which makes the robustness conclusions mainly established under medium-to-high overlap, lacking quantitative evidence for low overlap (e.g., <20% or 10–30%).
7. Insufficient evidence for AP-5D efficiency: The authors claim that AP-5D significantly alleviates the computational/memory burden of 5D correlation modeling, but the manuscript only provides big-O complexity derivations in the appendix and overall method-level comparisons in “seconds per sample + parameter count,” along with a module-level parameter breakdown (AP-5D about 9.44K parameters); however, it does not provide any comparison against a “naïve 5D convolution” implementation (runtime, peak memory, FLOPs/throughput curves, or accuracy trade-offs), and the existing ablations do not include a “naïve 5D” column, which leaves AP-5D’s efficiency and benefits without empirical support.

**Questions:**

Refer to weakness

---

### Official Review · Reviewer_mP9C · 2025-10-30

**Soundness:** 2
**Presentation:** 2
**Contribution:** 2
**Rating:** 4
**Confidence:** 5

**Summary:**

In this paper, the authors propose Dual-view Matching Aggregation (DuMA), an image-to-point cloud registration framework that harmonizes 2D-3D and 3D-3D correspondences for cross-modal alignment. By introducing a neighborhood-aware score aggregation module and an innovative Anchor-Pivot 5D encoder, DuMA enhances geometric consistency and achieves robust performance in complex scenes.

**Strengths:**

The proposed DuMA achieves SOTA performances on RGB-D Scenes V2 and 7Scenes datasets.

**Weaknesses:**

(1) The paper is difficult to follow, and its overall organization requires improvement.

(2) The proposed method represents only an incremental improvement over FreeReg [1], offering limited originality.

(3) The approach heavily depends on the accuracy of depth estimation and tends to perform poorly in challenging scenarios such as textureless regions.

(4) In Lines 71–73, the authors state that “we design a score aggregation module that fuses dual correspondence scores through a detailed analysis of neighborhood relationships, inducing a robust geometric verification effect and enforcing spatial consistency.” However, the paper does not provide a corresponding section or explanation detailing this analysis.

(5) In my opinion, the proposed Score Aggregation Module performs only a linear fusion of 2D-3D and 3D-3D matching scores. Such a simple interaction is insufficient to justify the claim that “unlike traditional methods that rely solely on feature similarity, our module leverages spatial relationships and geometric constraints to filter out ambiguous or incorrect matches.”

(6) The content of Figure 3 appears inconsistent with the description in Section 3.4, particularly regarding the dimensions of the feature maps.

(7) The paper lacks a comparison with Diff2I2P [2] in terms of runtime and model parameters. Please include this analysis.

(8) The authors claim in the Introduction that the proposed method outperforms FreeReg [1] in complex or cluttered environments; however, no dedicated experimental results are provided to substantiate this statement.

[1] Haiping Wang, Yuan Liu, Bing Wang, Yujing Sun, Zhen Dong, Wenping Wang, and Bisheng Yang. Freereg: Image-to-point cloud registration leveraging pretrained diffusion models and monocular depth estimators. In ICLR, 2024.

[2] Mu, Juncheng, et al. "Diff2I2P: Differentiable Image-to-Point Cloud Registration with Diffusion Prior." Proceedings of the IEEE/CVF International Conference on Computer Vision. 2025.

**Questions:**

(1) What are the concrete technical innovations that distinguish it from FreeReg [1]?

(2) How exactly are spatial relationships modeled in this module?

(3) In Lines 71–73, the authors mention a “detailed analysis of neighborhood relationships,” but no such analysis is presented in the paper. Could the authors provide the corresponding explanation or section?

(4) The experimental results do not include comparisons with Diff2I2P [2] in terms of runtime and model parameters. Could the authors provide this comparison?

(5) Could the authors include additional experiments or evidence to justify this claim?

---

### Official Review · Reviewer_7GFQ · 2025-10-31

**Soundness:** 3
**Presentation:** 3
**Contribution:** 2
**Rating:** 6
**Confidence:** 5

**Summary:**

The paper proposes DuMA, a novel framework for image-to-point cloud (I2P) registration. Its main innovation is the Anchor-Pivot 5D encoder for efficient high-dimensional feature fusion. The approach achieves state-of-the-art (SOTA) results on multiple datasets (RGB-D Scenes V2, 7Scenes, KITTI-DC).However the integration of geometric and visual cues is not quite innovative, which has been adopted by prior methods.

**Strengths:**

1. Evaluation Results
DuMA outperforms prior SOTA methods by a good percentage on average, around 5-10. But compared to the improvement of Diff-Reg to his priors, the improvement is still limited.
However, on most/many scenes, DuMA does not have the best accuracy.
Ablation studies (fusion weight, SAM module, modality contributions) are well-designed and provide clear insights.

2. Writing
Clear and easy to follow

**Weaknesses:**

1. Novelty. I believe the main novelty of this comes from the  Anchor-Pivot 5D encoder for efficient high-dimensional feature fusion, which is relatively limited. Taking advantage of visual and geometric clues is explored by prior methods already,

2. Dependency on Depth Estimation Quality
The model depends heavily on monocular depth (Zoe-Depth). The paper addresses this limitation only briefly in the conclusion; an explicit quantitative analysis of how noisy depth affects registration would strengthen the claim of robustness.

3. Despite the efficient high-dimensional feature fusion module, design, the inference time is still slow and relatively similar to Diff-Reg. In the  runtime analysis, the authors only show result of two baselines. How about the other baselines?

**Questions:**

1. The Diff-Reg claims that they use scene 11-14 for validation. But you use 11-14 for testing. There are some bias here to clarify.

2. Table 9 typo: DuSA-Reg

3. I personally think the novelty of the paper is limited. but the results seem good, around 5-10 point increases on average.  I would like to hear from other reviewers whether such an improvement is significant.

4. There are traces that LLM  model is used to refine the grammar at least. But they do not claim in the paper.

---

### Official Review · Reviewer_noE2 · 2025-11-03

**Soundness:** 3
**Presentation:** 3
**Contribution:** 3
**Rating:** 6
**Confidence:** 4

**Summary:**

The paper proposes DuMA, a new framework for image-to-point cloud (I2P) registration that leverages dual-view matching between 2D images and 3D point clouds. It combines 2D-3D image patches to point clusters matching and point cloud to point cloud matching. The matchings are achieved by a Score Aggregation Module (SAM) that uses an Anchor-Pivot 5D Encoder to aggregate high-dimensional matching scores while enforcing geometric consistency.

**Strengths:**

The paper combines both 2D–3D and 3D–3D matching to leverage complementary visual and geometric cues and improves robustness in complex or ambiguous scenes.

The Anchor-Pivot 5D Encoder decomposes 5D matching scores into separate 2D and 3D processing branches, reduces computational complexity and memory usage compared to a naive 5D convolution, and enhances geometric consistency by modeling local spatial relationships.

The method has improvements over other methods for indoor and outdoor datasets.

Ablations are comprehensive.

**Weaknesses:**

The method relies on monocular depth estimators (e.g., Zoe-Depth) to lift 2D images to 3D, which means its performance degrades with poor depth estimation, especially in textureless or reflective regions, and it also limit the new contributions of the paper.

It seems the method struggles in textureless regions or when both visual and geometric cues are weak. It may also fail in cases with severe scale ambiguity or sparse point clouds. Please discuss.

The time efficiency: although the method is more efficient than a full 5D convolution, the Anchor-Pivot 5D Encoder still adds complexity. The method has a slightly slower inference time compared to some baselines (e.g., Diff-Reg), which proves it.

There is no explicit depth uncertainty modeling, what if the depth is not accurate?

**Questions:**

See above.

---

### Note · Authors · 2025-11-13

I have read and agree with the venue's withdrawal policy on behalf of myself and my co-authors.